# Best Practices for the Use of High-Frequency Ultrasound to Guide Aesthetic Filler Injections—Part 2: Middle Third of the Face, Nose, and Tear Troughs

**DOI:** 10.3390/diagnostics14222544

**Published:** 2024-11-13

**Authors:** Roberta Vasconcelos-Berg, Stella Desyatnikova, Paula Bonavia, Maria Cristina Chammas, Alexander Navarini, Rosa Sigrist

**Affiliations:** 1Margarethenklinik-University Hospital of Basel, CH-4051 Basel, Switzerland; paulavalentina.bonavia@usb.ch (P.B.); alexander.navarini@usb.ch (A.N.); 2The Stella Center for Facial Plastic Surgery, Seattle, WA 98101, USA; stella@doctorstella.com; 3Department of Radiology, School of Medicine, University of São Paulo, São Paulo 05403-010, Brazil; mcchammas@hotmail.com (M.C.C.); rm@sigrist.com.br (R.S.)

**Keywords:** filler, hyaluronic acid, guided injections, nose, tear troughs, nasolabial fold, zygomatic, preauricular, Doppler ultrasound

## Abstract

The midface is a key area in facial aesthetics, highly susceptible to age-related changes such as fat pad absorption, bone resorption, and loss of skin elasticity. These changes lead to the formation of prominent folds, such as the nasolabial fold. In addition, critical vascular structures and non-vascular components, such as the facial artery, angular artery, and parotid gland, make this region prone to complications during filler injections. High-frequency ultrasound (HFUS) offers real-time, radiation-free visualization of facial anatomy, enabling injectors to accurately target the desired treatment planes and avoid critical structures. This article is the second in a series of articles on ultrasound-guided facial injections and focuses on the midface. It provides a detailed overview of the sonographic anatomy of key areas, including the nose, tear trough, nasolabial fold, zygomatic, and preauricular regions. Step-by-step techniques for ultrasound-guided filler injections are described, emphasizing the importance of scanning both before and during injections to ensure safe filler placement. By using ultrasound in this area, injectors can possibly minimize risks such as vascular occlusion and other complications, such as the Tyndall effect and intra-parotid injection. With ongoing advancements, ultrasound-guided injections are expected to become more refined, enhancing both aesthetic outcomes and patient safety.

## 1. Introduction

The midface is a region of significant importance in the aesthetic composition of the face. It is also highly affected by the aging process, with notable fat pad absorption in the medial malar and subocular regions, along with the loss of collagen in the dermis, ligaments, and SMAS (superficial musculoaponeurotic system). These lead to a decrease in tissue support. The soft tissue losses, when combined with bone resorption, contribute to the formation of prominent folds in this area, such as the nasolabial fold.

The nose is a key feature of this region and has increasingly become the focus of minimally invasive aesthetic procedures in recent years. Several critical anatomical structures are located in the midface, including the nasal and central facial vessels. Additionally, when performing aesthetic injections in the preauricular region, it is important to consider the presence of the parotid gland and the masseter muscle.

As discussed in the first article of this series [1], where we covered ultrasound-guided filler injections in the upper third of the face, high-frequency ultrasound (HFUS) is a radiation-free imaging technique that offers real-time visualization of facial anatomical structures, ensuring the correct injection plane is accurately achieved.

Continuing this series of articles, we aim to provide a comprehensive overview of the ultrasonographic anatomy of the midface and offer detailed, reproducible ultrasound-guided techniques for the nasal, tear trough and medial malar, nasolabial fold, zygomatic, and preauricular regions.

## 2. Materials and Methods

This narrative review draws upon the authors’ extensive experience with high-frequency DUS-guided injections in the midface. The techniques and recommendations discussed in this paper reflect the authors’ personal perspective and should be interpreted as such.

The ultrasound devices utilized in this study included a LOGIQ E10 and LOGIQ e (GE Healthcare, Waukesha, WI, USA) equipped with linear probes of 6–24 MHz and 8–18 MHz, a Venue Fit (GE Healthcare, Waukesha, WI, USA) featuring a linear probe of 4–20 MHz, and an ACUSON Sequoia (Siemens Medical Solutions, Mountain View, CA, USA) with a linear probe of 6–18 MHz.

Before the procedures, the treatment area was disinfected, and the linear probe was covered with a sterile transparent dressing, Opsite^®^ (Smith & Nephew Medical, Suzhou, China), following a technique previously described by our team [2].

The injections were performed either with a blunt 22 G or 25 G cannula or a 27 G or 30 G needle.

As previously recommended by this group [2], we classify the guided injection methods into the following didactic approaches:**Scan before injecting:** The treatment area is scanned immediately prior to the injection to identify the location of the vessels. The position of the main arteries can be marked on the patient’s skin.**Scan while injecting:** In this approach, the cannula is visualized in real-time and guided to the target anatomical region, ensuring vascular structures are avoided.

Both in-plane and out-of-plane techniques can be employed to position the ultrasound probe relative to the cannula. The in-plane technique is generally preferred, as it allows real-time visualization of the cannula’s long axis. However, if the cannula is positioned too close to a vessel, the out-of-plane technique can be used to confirm that the cannula is not within the vessel.

## 3. Results

### 3.1. Nose

#### 3.1.1. Sonographic Anatomy of the Nose

The sonographic examination of the nasal region provides a comprehensive layer-by-layer evaluation of its anatomical structures, progressing from superficial to deep [3] (Figure 1a). The skin is the most superficial layer visualized and it shows an increased thickness at the nasal tip. Underlying the skin, the hyperechoic fibroadipose tissue of the hypodermis has less prominent fatty lobules. The procerus muscle is identified at the radix, while the nasalis muscle extends along the nasal dorsum. The nasal bones appear as hyperechoic linear structures. The upper and lower lateral cartilages are visualized as hypoechoic convex formations. A distinct linear hyperechoic interface beneath the cartilage indicates the presence of air within the nasal cavity, further aiding in the anatomical delineation.

The nasal region’s vascularization consists of a complex network of blood vessels originating from both the internal and external carotid arteries, resulting in extensive anastomoses. The primary arterial supplies to the nose are the dorsal nasal arteries, which have been described to be positioned parallel to the midline in 38% of cases [4] (Figure 1b), the external nasal arteries, and all branches of the internal carotid arteries. The lateral nasal arteries positioned at the nasal alae and the columellar arteries are all branches of the external carotid arteries. Identifying these arteries can be challenging due to anatomical variations that may affect their typical presentation [4]. While cadaveric studies have shown the vessels to run in the fibromuscular layer [5], ultrasound studies have identified some arteries in the supraperiosteal plane [6,7]. Given these variations, ultrasound should be utilized as a reliable tool to accurately locate and assess the course of these vessels.

#### 3.1.2. Ultrasound-Guided Filling Techniques of the Nose

Technique 1: Deep injections with a needle

The majority of the nonsurgical rhinoplasty cases are suitable for this type of filler injection. Our technique places small droplets of high G’ hyaluronic acid filler in the supraperiosteal/supraperichondrial plane.

The injection sites are selected in various regions of the nose, including the radix, tip, supratip, dorsum, and nasal spine. Prior to injection, these areas are scanned using Doppler ultrasound (“scan before injecting”) to assess the presence and location of blood vessels. Once the safer injection points are identified, the filler is administered in small aliquots using a 29 G or 31 G needle, placed in a deep supraperiosteal or supraperichondrial plane. Aspiration is performed when injections are made near blood vessels to further enhance safety.

Technique 2: Deep injections with a cannula and ultrasound-guided filler placement

This technique can be employed when Doppler scanning prior to injection reveals the presence of blood vessels in the intended treatment area. Additionally, it is a viable option for injectors who prefer using cannulas over needles. A blunt-tip cannula, 25 G or larger, is utilized.

For the treatment of the dorsum, the cannula can be inserted from the tip in a cephalic direction, or from the glabella in a caudal direction to target the deep radix. For columella injection, the cannula is inserted starting from the nasal tip.

Once the cannula is correctly positioned in the deep tissue plane, an ultrasound probe, with a small amount of sterile gel, is placed along the cannula (in-plane technique). After confirming the cannula tip is in the correct position, away from blood vessels and within the appropriate tissue plane (Figure 2), the filler is injected slowly in a linear retrograde fashion under ultrasound guidance (Figure 3a–c).

### 3.2. Tear Trough and Malar Medial Region

#### 3.2.1. Sonographic Anatomy of the Tear Trough and Malar Medial Region

The medial infraorbital region, commonly referred to as the tear trough, can be visualized sonographically (Figure 4), with its three fascial layers clearly delineated in its cranial part: the skin, the orbicularis oculi muscle (OOM), and the periosteum [8]. The primary vascular structures in this area include the angular vein, a prominent superficial vessel that penetrates the OOM [8]. Additionally, an extension of the angular artery (Figure 5) could be observed within the tear trough and was recognized when performing ultrasound [9].

The infraorbital foramen, which gives rise to the infraorbital vessels and nerve, is located in the medial malar region, also denominated mid-cheek, and can be recognized as a gap in the bone during ultrasound evaluation (Figure 4 and Figure 5). It should be avoided when performing supraperiosteal injections.

#### 3.2.2. Ultrasound-Guided Filling Techniques of the Tear Trough and Malar Medial Region

The aging process of the subocular region, including the tear trough, is complex and involves not only volumetric loss due to fat compartment absorption and bone resorption but also the loss of elasticity in retaining ligaments, skin atrophy, and pigmentary changes [10]. Additionally, with aging, lymphatic drainage in the region is considered less efficient, contributing to a greater propensity for local edema. Hyaluronic acid fillers can restore volume and provide structural support to this region.

In many cases, filling adjacent areas such as the zygomatic and medial malar regions reduces or eliminates the need for a tear trough filler. This occurs due to the repositioning of the ligamentous structures and, consequently, the fat pads. This is an effective strategy to avoid using large volumes of hyaluronic acid in the tear trough.

The medial malar region can also be treated with fillers in the context of facial aging, providing support for the tear trough correction. Additionally, this area often exhibits significant volume loss in high-performance athletes or patients who have undergone intense weight loss. Furthermore, this region contains the malar eminence, which can be enhanced with fillers in women to achieve greater facial beautification.

Tear trough

Although there are also techniques for filling this region using needles, due to their high-risk nature, the safest technique is the use of a blunt 25 G, or larger, cannula. The filler chosen should be low in hydrophilicity to minimize changes in the final result post treatment. Under-correction is advised, as post-treatment water absorption, particularly in patients prone to local edema, can lead to periorbital swelling.

Given that the skin in this area is very thin and there is no subcutaneous fat in the medial tear trough, the filler should be placed as deeply as possible, preferably in the supraperiosteal plane, to avoid the Tyndall effect.

The vessels of greatest concern are the angular vein, as its compression may contribute to periorbital edema [11], and the angular artery near the nose in the medial tear trough. Occlusion of the latter can lead to skin necrosis, and in cases of distant embolism, blindness.

Filling can be performed in either the medial or lateral tear trough. In both cases, we recommend using the “scan while injecting” technique (Figure 6a,b). If the cannula is visualized in close proximity to any vascular structures, it should be repositioned. The cannula’s position relative to the tissues can also be assessed and the depth adjusted as needed. Figure 6b illustrates an appropriate cannula position, close to the periosteum and away from vascular structures.

Another possible approach for treating this area is the needle injection technique. A 29 G or 30 G needle is used to deliver micro-aliquots of high G′, low-hydrophilicity HA filler to the supraperiosteal plane, positioned below the tear trough ligament and above the infraorbital foramen. The injection area should be marked and scanned prior to the procedure (“scan before injecting”). Care must be taken to avoid the angular vein, angular artery, and the infraorbital neurovascular bundle. The needle is inserted at the desired point, perpendicular to the skin and bone surfaces.

Malar medial

Filling in this region can be performed with either a needle or a blunt cannula. Special care should be taken to avoid injecting near the infraorbital foramen or into any vessels.

Needle injection at 90 degrees: The authors primarily use this technique at the point of the malar eminence. This point is marked on the skin at the intersection of the lines connecting the lateral canthus to the oral commissure and the mid-tragus to the nasal ala (Figure 7). The area is then scanned with ultrasound (“scan before injecting”) to check for the presence of vessels or the foramen in the deep (periosteal) plane. If neither is present, a slow injection is performed following aspiration.

Blunt cannula injection: To volumize the region, our preferred approach is the use of a cannula, which is kept in the deep subcutaneous plane. While the risk of occlusion is lower, there is still the possibility of compressing the infraorbital foramen with the filler, often resulting in persistent pain or paresthesia. A good strategy is to use ultrasound to identify the foramen in advance (Figure 8), mark it on the skin, and avoid this area during the cannula injection. The “scan while injecting” technique can also be used when the cannula is very close to the marked location of the foramen on the skin, to ensure that no injection occurs near it.

### 3.3. Nasolabial Fold

#### 3.3.1. Sonographic Anatomy of the Nasolabial Fold

The sonographic examination of the nasolabial fold provides a detailed evaluation of its anatomical structures, which is crucial for ensuring safe and precise filler injections (Figure 9a). Beneath the epidermis and dermis, the subcutaneous tissue appears hypoechoic. Following this layer is the muscular layer, primarily composed of the levator labii superioris alaeque nasi and levator anguli oris muscles and visualized as distinct hypoechoic structures beneath the subcutaneous tissue. Of particular importance during filler injections is the angular artery, whose course can vary significantly in both depth and position, sometimes located laterally or medially to the nasolabial fold, as demonstrated in previous ultrasound studies [12]. The artery can even change planes, as shown in Figure 9b. Since the cannula is typically aligned with the artery during the procedure, there is an increased risk of inadvertently cannulating the vessel and causing an occlusion; therefore, ultrasound guidance is crucial for precisely identifying the artery’s location and reducing the risk of vascular injury.

#### 3.3.2. Ultrasound-Guided Filling Techniques of the Nasolabial Region

For ultrasound-guided filler injections in the nasolabial fold and piriform fossa, our approach is based on techniques traditionally used to treat this area without direct visualization [13,14]. Our group proposes the following approaches:Technique 1: Subcutaneous filler placement with a blunt cannula

The target area with this approach is the nasolabial fold itself, aiming to soften the wrinkle commonly formed between the nose and the corner of the mouth. As this plane is well-suited for cannula use, we utilize the “scan while injecting” technique.

The procedure begins with the insertion of the cannula into the skin and subcutaneous tissue. The entry point is typically located 1 to 1.5 cm lateral to the oral commissure, following the natural wrinkle line. A 25 G or larger blunt-tip cannula is used. The ultrasound probe is then positioned along the cannula’s trajectory (in-plane technique), with a small amount of sterile gel applied for conductivity (Figure 10). The operator confirms that the cannula is correctly positioned within the desired layer, ensuring that its tip is not in contact with any branches of the facial artery or vein. Once the correct position is verified, the filler is slowly injected retrogradely, under ultrasound guidance, from the nasal to the labial angle.

Technique 2: Supraperiosteal filler placement with a needle

The goal of this technique is to restore lost volume in the piriform fossa region. Although this area can also be treated using a blunt cannula, as described for the nasolabial fold, we outline below the needle technique, which involves placing the filler in the supraperiosteal plane within the piriform fossa by inserting the needle perpendicularly to the skin.

The injection point is located 0.3 to 0.5 cm lateral to the nasal ala. This approach is recommended for addressing severe volume loss in the piriform fossa, which helps to elevate the entire nasolabial fold. Initially, the injection site should be scanned with Doppler ultrasound (“scan before injecting”) (Figure 9) to rule out the presence of the facial artery. If the artery is found at this point, the injection site should be moved, and the procedure is repeated until a safe location is identified. Once the injection site is determined, the filler is injected slowly with a 27 G needle at a 90-degree angle in the supraperiosteal plane after aspiration.

### 3.4. Zygomatic and Preauricular Region

#### 3.4.1. Sonographic Anatomy of the Zygomatic and Preauricular Region

The sonographic anatomy of the zygomatic and preauricular regions consists of complex, multi-layered structures that are essential for safe and effective filler injections. In the zygomatic region, the use of ultrasound provides a clear visualization of the skin, subcutaneous tissue, and the superficial musculoaponeurotic system (SMAS), followed by the zygomatic muscles and bone (Figure 11a,b). The SMAS is a critical landmark that separates the superficial and deep fat compartments, helping to guide the injector to avoid placing filler in this layer, which could lead to complications such as nodule formation [15]. The zygomaticofacial foramen (Figure 11b) and any additional accessory foramina can be recognized and avoided.

In the preauricular region, the sonographic anatomy is more complex due to the proximity of the parotid gland. The layers visible on ultrasound (Figure 12a–d) include the skin, subcutaneous fat, SMAS, transverse facial artery, parotid fascia, parotid gland, and masseter muscle [16]. The subcutaneous tissue is particularly thin in this area, often measuring between 2 and 4 mm, which increases the risk of inadvertently injecting filler into the parotid gland. This can lead to serious complications, including inflammation, abscess formation, and chronic parotitis [16]. Even if the filler is placed near the glands, there is still a chance of developing inflammation [17]. Additionally, anatomical variations such as accessory parotid glands (Figure 12d) or prominent anterior prolongations (Figure 12a) that cover the upper third of the masseter muscle should be carefully avoided to prevent these issues [16].

#### 3.4.2. Ultrasound-Guided Filling Techniques of the Zygomatic and Preauricular Region

Zygomatic Region

Hyaluronic acid filler injections in the zygomatic region are common, whether to restore volume lost in the subcutaneous tissue due to aging or to create a lifting effect. This lifting is achieved by elevating the zygomatic ligament (Figure 11b) and pulling back the SMAS, caused by supraperiosteal deposits of hyaluronic acid on the zygomatic bone.

Technique 1: Supraperiosteal filler placement

This can be performed with either a needle or a blunt cannula. In the needle technique most frequently described [18], the following three points are marked: the first on the zygomatic suture and the other two 0.5 cm equidistant from it, anteriorly and posteriorly, following the zygomatic bone (Figure 13).

Since the injection is performed perpendicularly with a needle, these points are scanned beforehand using ultrasound (“scan before injecting”) to rule out the presence of the zygomaticofacial foramen (Figure 11b), the transverse facial artery (Figure 12c), and any of its branches. Then, hyaluronic acid is injected deeply at a 90-degree angle to the skin using a 27 G needle. Typically, 0.1 to 0.2 mL of filler is injected per point slowly following aspiration.

The same area can also be injected supraperiosteally using a 25 G or larger blunt cannula. In this case, ultrasound can be used during the injection (“scan while injecting”) to check the position of the cannula near the periosteum and reposition it if necessary (Figure 14). Additionally, it helps to ensure that there are no blood vessels in contact with the cannula.

Technique 2: Subcutaneous filler injection

In cases of significant volume loss in the zygomatic region, it may be desirable to inject the filler into the deep subcutaneous plane using a blunt cannula. Ultrasound can assist in precisely locating the desired plane. As with the supraperiosteal plane, it can also help avoid contact between the cannula and the transverse facial artery. The technique used in this case is “scan while injecting” (Figure 15).

Preauricular Region

In this region, we recommend using a blunt cannula 25 G or larger in the superficial subcutaneous plane. Hyaluronic acid will be applied in a fanning technique, and the entry point can be made in the posterior zygomatic region, the mandibular angle, or the jowl area, depending on the injector’s preference.

The “scan while injecting” technique should be used in this region, which will constantly assess the cannula’s position to ensure that it is outside the parotid gland and above the parotid masseteric fascia (Figure 16). Each time the direction of the cannula is changed, its position should be reassessed.

## 4. Discussion

This article follows the first in the series, which covered the upper third of the face [1], and provides clear and reproducible methods for approaching each facial region to perform ultrasound-guided filler injections.

The midface is a commonly targeted area for aesthetic fillers, whether in its posterior portion behind the ligament line, to reposition anatomical structures before addressing the central face, or to volumize areas affected by age-related volume loss, such as the tear trough, medial malar region, and piriform fossa. However, the midface contains numerous vascular structures, including the facial artery near the nasolabial fold, the angular arteries and veins, the infraorbital vessels in the tear trough area, the transverse facial artery, and the zygomatico-orbital artery in the zygomatic region. Additionally, non-vascular structures such as the parotid gland and the infraorbital nerve are key components of the region’s anatomy. Inadvertent hyaluronic acid injections in these areas can lead to complications of varying severity.

Nasal filler injection remains one of the highest risk areas for blindness, as described in the recent update by Doyon et al. [19]. Nasal ultrasound anatomy and vascular anatomy have been described in multiple publications [20,21,22,23,24].

Nevertheless, nasal vascular anatomy can be extremely variable and the use of high-resolution ultrasound is advocated for nonsurgical rhinoplasty using filler. Ultrasound-guided hyaluronic acid filler techniques for the nasal region have been described by Lee [7], Lee [5], Bravo [25], and Yi [26].

The tear trough is also an important aesthetic unit that can be treated with fillers. Noell et al. [9] proposed the use of angular artery identification in the tear trough region prior to filler injections, utilizing a 15 MHz linear Doppler transducer to avoid intravascular complications. In the same year, the flow of the angular vein was studied [8] in both the static position and during facial movement using ultrasound in volunteers scheduled for filler treatment. The vein was identified within the orbicularis oculi muscle in 100% of cases, and no flow was detected during smiling.

Recently, a few authors have studied the nasolabial fold (NLF) region using high-frequency Doppler ultrasound to determine its relationship with the facial artery (FA). Lee et al. [5] evaluated 40 patients (80 NLFs) and found that in 69% of cases, the FA ran directly beneath the fold. Of these, in the majority of cases (76%, 42 out of 55), the artery was located at an intermediate depth, either in the subcutaneous or intramuscular plane. In 18% of cases, the artery was more superficial and situated in the subdermal plane, while in 5% of cases, it was deeper and situated within the deep medial buccal fat layer or supraperiosteal plane. This article proposes that the artery’s location should be determined prior to the procedure, allowing hyaluronic acid to be injected accordingly to avoid the artery. A similar technique is suggested by Boz et al. [27], whose study observed a more asymmetrical course of the artery in females compared to males. Another interesting study [12] assessed 184 hemifaces using ultrasonography and observed significant variability in the depth of the facial artery along its course and between individuals, concluding that there is no universally safe depth for injection in the nasolabial fold.

Ultrasound-guided zygomatic region filler injections have not been extensively explored in the medical literature. However, Bravo et al. [28] demonstrated, using high-frequency ultrasound, the supraperiosteal injection of hyaluronic acid on the zygomatic bone in a patient, performed with both needle and cannula techniques. In this case, ultrasound was not used to identify anatomical or vascular structures but rather to document the injection within the desired plane. Similarly, Li et al. [18] describe the use of ultrasound to ensure injection into the periosteal plane, utilizing a needle.

The preauricular region has been increasingly explored for the use of hyaluronic acid in recent years, both to create a more angular facial appearance, particularly in facial masculinization procedures, and to promote a lifting effect, especially when combined with zygomatic filler and applied posterior to the ligamentous line [29]. However, the presence of the parotid gland just beneath the skin makes injections in this area susceptible to complications. Schelke et al. [16] described two cases of complications following intraparotid injections. Additionally, in a video demonstrating an accidental intraparotid injection, the authors showed that the patient did not report any pain, and the injector did not perceive any difference in tissue texture during the injection.

Another key benefit of using ultrasound to guide midface filler injections is the ability to accurately select the correct treatment plane, ensuring that the filler is placed at the desired depth. In this article, for instance, we illustrate the use of ultrasound in the tear trough, targeting the supraperiosteal plane. Injecting the filler at this deeper level helps minimize complications such as the Tyndall effect and regional edema. We anticipate that in the coming years, ultrasound-guided injection techniques will continue to evolve, becoming increasingly refined and precise, allowing for specific anatomical regions to be targeted to achieve different aesthetic outcomes or reduce complications.

## 5. Conclusions

Ultrasound-guided techniques for midface filler injections can be an effective tool for reducing complications and improving the aesthetic outcomes of treatments. These guidelines provide key concepts for injectors who wish to integrate this technology into their daily practice. By adopting these approaches, practitioners can achieve safer, more precise results, ultimately enhancing patient satisfaction and treatment success.

## Figures and Tables

**Figure 1 diagnostics-14-02544-f001:**
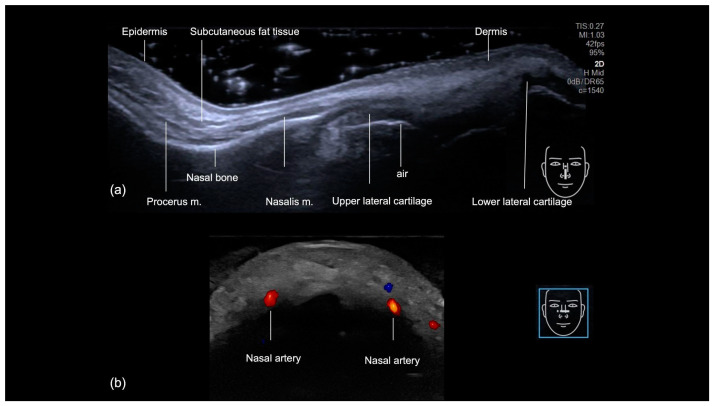
US of the nose. (**a**) Grayscale US at 18 MHz, showing the layers in the longitudinal view. (**b**) Color Doppler US at 20 MHz, demonstrating the nasal arteries in the transverse view (in red).

**Figure 2 diagnostics-14-02544-f002:**
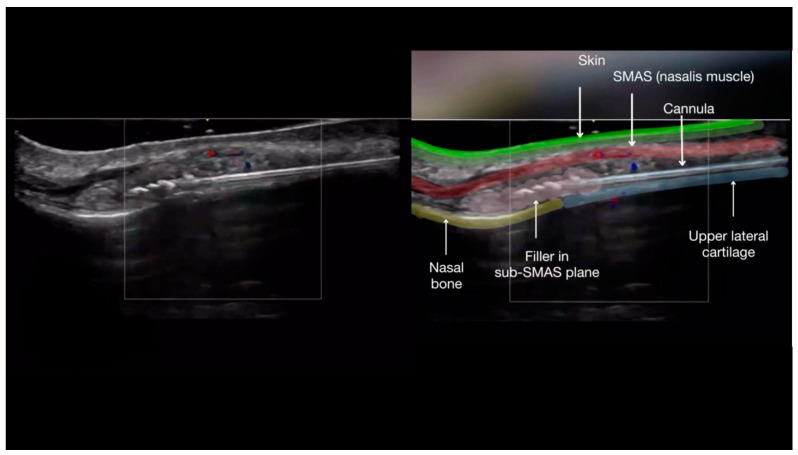
Color Doppler US of the nose with a 20 MHz probe showing the layers of the nose while injecting filler.

**Figure 3 diagnostics-14-02544-f003:**
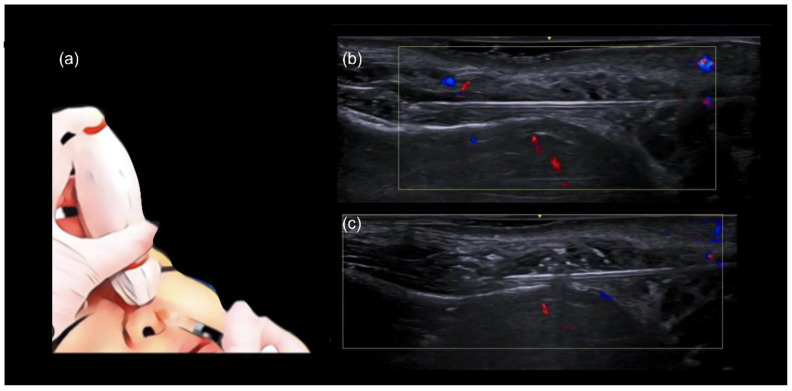
Scan while injecting in the nose. (**a**–**c**) Color Doppler US with a 20 MHz probe demonstrates the in-plane technique in the dorsum, which allows real-time visualization of the cannula as filler is being deposited in a linear retrograde fashion. (**a**) Injection technique using a blunt cannula 25 G. Here we demonstrate the” scan while injecting” technique in the nose. (**b**) Plane of injection is checked for vessels. (**c**) Filler starts to be deposited.

**Figure 4 diagnostics-14-02544-f004:**
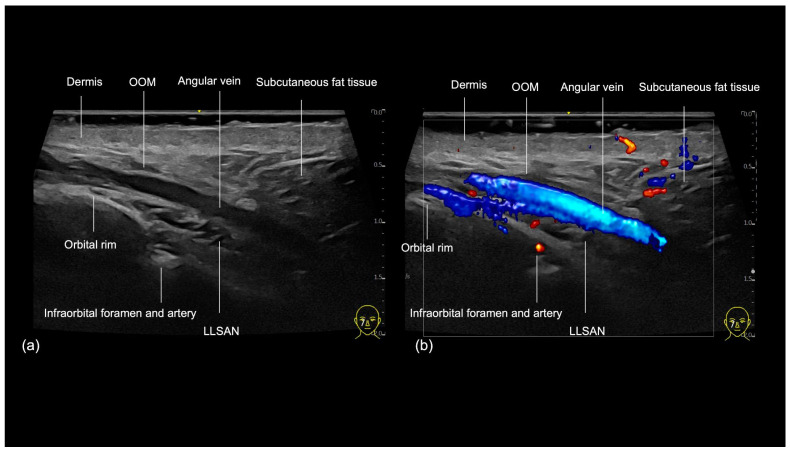
US of layers in the tear trough (oblique view). (**a**) Grayscale. (**b**) Color Doppler US at 20 MHz. OOM: orbicularis oculi muscle. LLSAN: levator labii superioris alaeque nasalis muscle.

**Figure 5 diagnostics-14-02544-f005:**
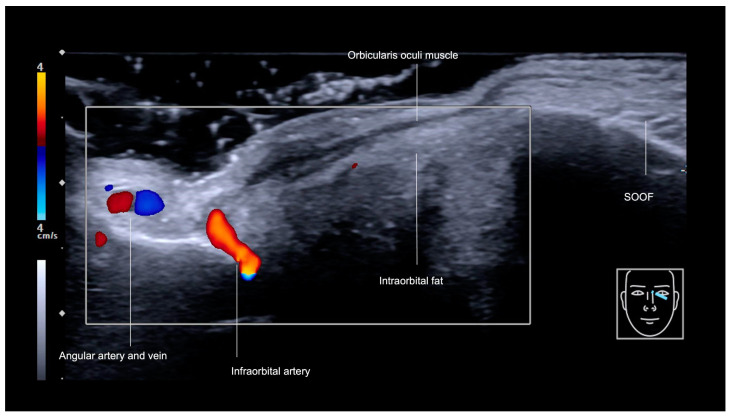
Color Doppler US of the tear trough (transverse view) at 18 MHz, showing the angular artery and vein, infraorbital artery, and SOOF.

**Figure 6 diagnostics-14-02544-f006:**
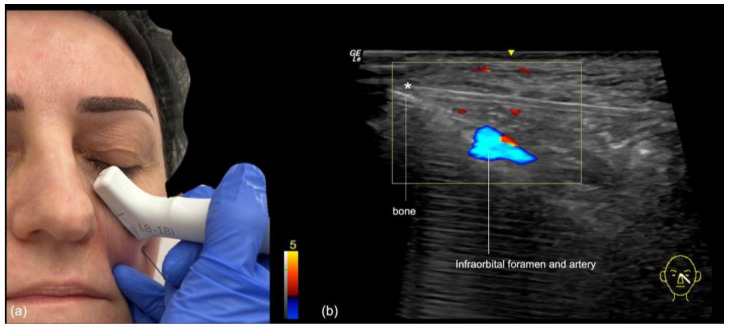
US-guided injection in the medial tear trough. (**a**) In-plane technique and scan while injecting. (**b**) Color Doppler US imaging showing cannula close to the bone at 18 MHz, and its tip (*) outside of vessels.

**Figure 7 diagnostics-14-02544-f007:**
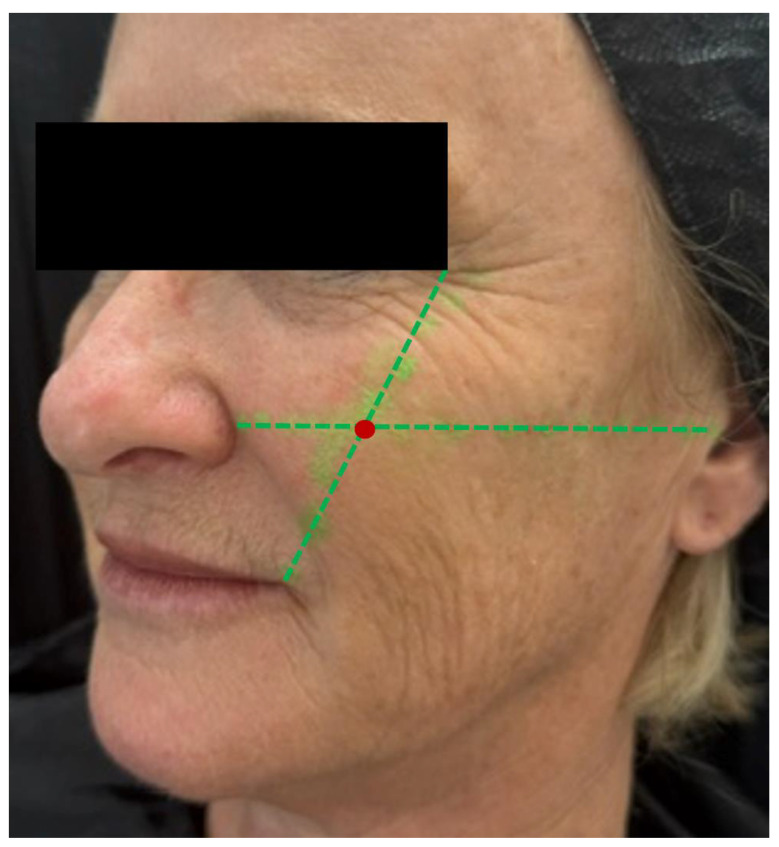
Marking of the malar eminence. The point is marked (red mark) on the skin at the intersection of two lines (green dotted lines): one connecting the lateral canthus to the oral commissure, and the other from the mid-tragus to the nasal ala. The area is then scanned using ultrasound (“scan before injecting”) to check for the presence of vessels or the foramen in the deep (periosteal) plane. If neither is detected, a slow injection is performed following careful aspiration.

**Figure 8 diagnostics-14-02544-f008:**
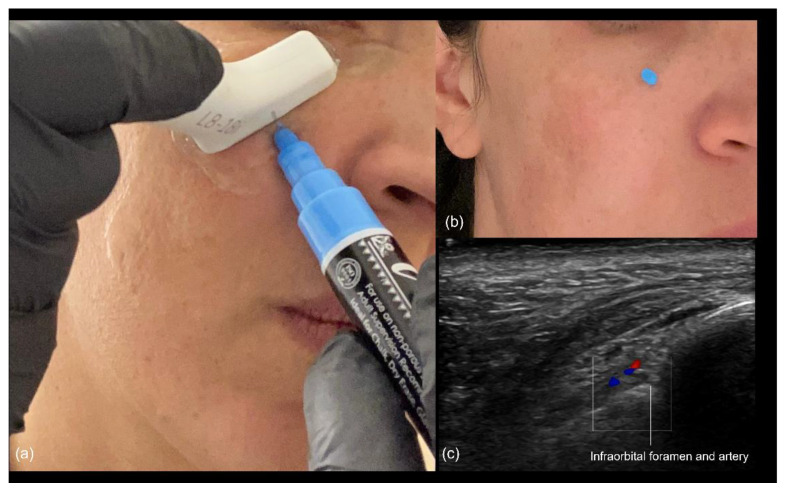
(**a**) Identification of the infraorbital foramen prior to injection in the medial malar region. (**b**) Appearance of the skin after marking the location. (**c**) Color Doppler US imaging showing the infraorbital foramen and artery at 18 MHz.

**Figure 9 diagnostics-14-02544-f009:**
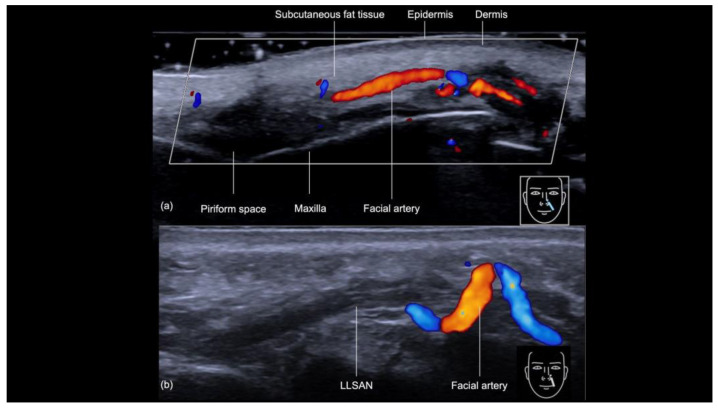
(**a**,**b**) Color Doppler US of the nasal labial fold at 18 MHz demonstrating its layers on the longitudinal view. The depth and course of the facial artery are clearly visible. Note the variation in the course of the facial artery (**b**), which can be torturous and located in different planes. LLSAN: Levator labii superioris alaeque nasi.

**Figure 10 diagnostics-14-02544-f010:**
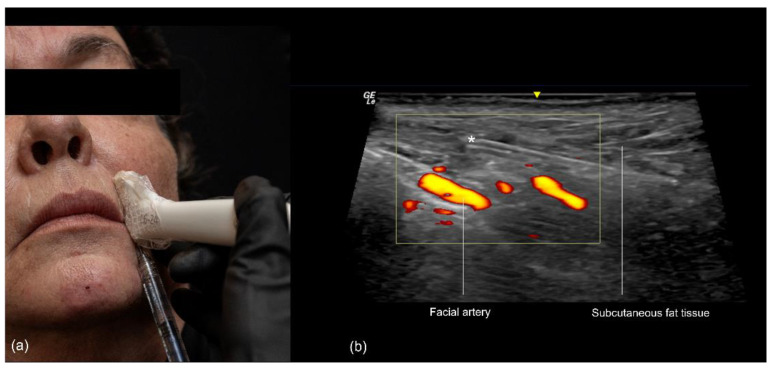
US-guided injection in the nasolabial fold (**a**) In-plane technique, scan while injecting. (**b**) Color Doppler US imaging showing the cannula in the subcutaneous fat tissue at 18 MHz, and its tip (*) outside of vessels.

**Figure 11 diagnostics-14-02544-f011:**
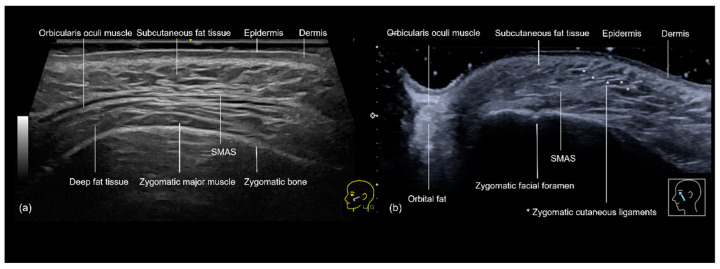
Grayscale ultrasonography of the zygomatic region showing its layers. (**a**) Transverse view at 20 MHz. (**b**) Longitudinal view) at 18 MHz. SMAS: superficial musculoaponeurotic system. * Zygomatic cutaneous ligaments.

**Figure 12 diagnostics-14-02544-f012:**
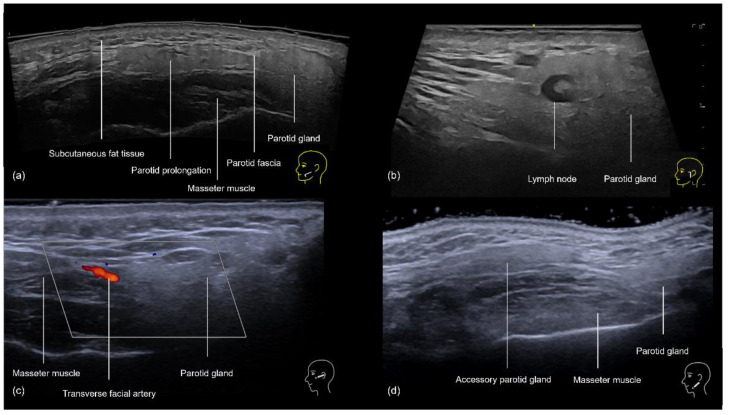
US of the preauricular region layers. (**a**) Prominent anterior prolongation of the parotid gland. (**b**) A normal lymph node inside the parotid gland with a hyperechoic hilum (transverse panoramic and longitudinal view at 18 MHz, respectively). (**c**) Color Doppler highlighting the transverse facial artery emerging between the masseter muscle and the parotid gland. (**d**) Grayscale US demonstrating an accessory parotid gland superficial to the masseter muscle (transverse view at 18 MHz).

**Figure 13 diagnostics-14-02544-f013:**
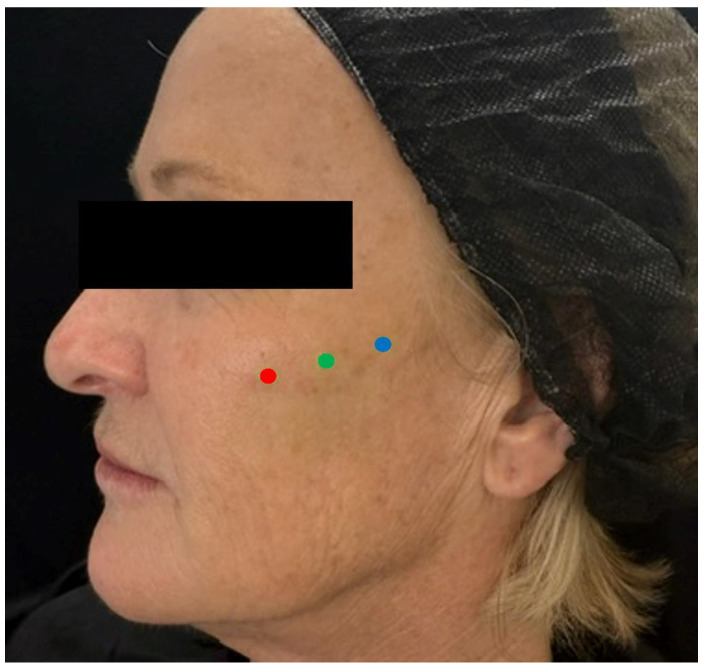
Points marked are as follows: zygomatic suture (green mark), 0.5 cm anteriorly (red mark), and 0.5 cm posteriorly (blue mark).

**Figure 14 diagnostics-14-02544-f014:**
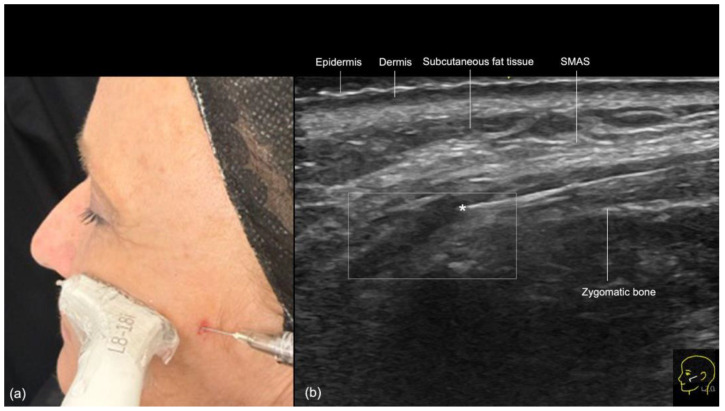
Filler treatment of the supraperiosteal plane in the zygomatic region with a 22 G cannula. (**a**) Using the “scan while injecting” technique, the depth of the cannula can be controlled, ensuring that the supraperiosteal plane is reached. (**b**) Contact with vascular structures can be ruled out using color Doppler US imaging which shows the cannula in the supraperiosteal plane at 18 MHz and its tip (*) outside of vessels.

**Figure 15 diagnostics-14-02544-f015:**
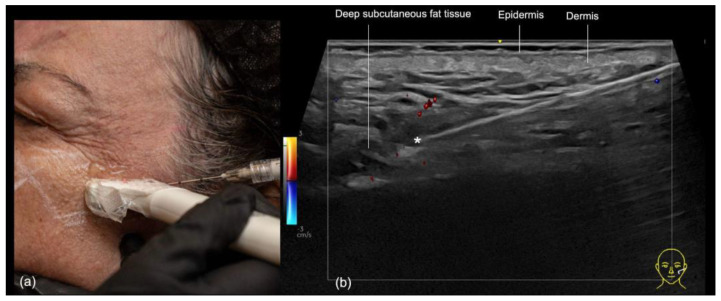
US-guided injection in the zygomatic region. (**a**) In-plane technique and scan while injecting. (**b**) Color Doppler US imaging showing the cannula in the deep subcutaneous plane at 24 MHz, and its tip (*) outside of vessels.

**Figure 16 diagnostics-14-02544-f016:**
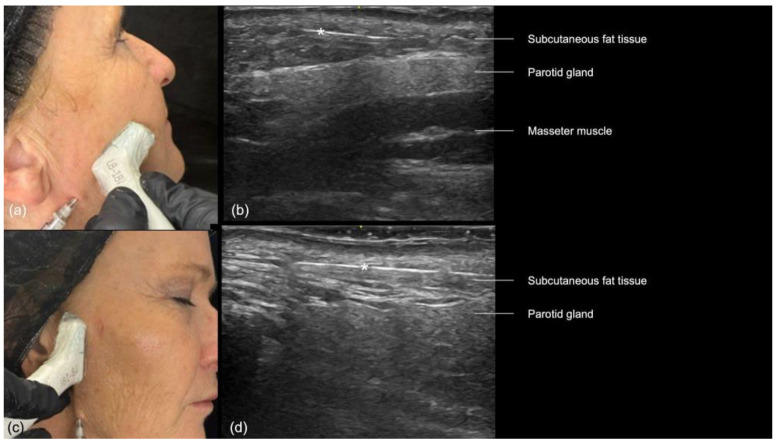
US-guided injection in the preauricular region. (**a**) Transverse. (**b**) Longitudinal injections. (**c**,**d**) B-mode US imaging showing the cannula (*) in the superficial subcutaneous plane, superficial to the parotid gland, at 18 MHz.

## Data Availability

No new data were created or analyzed in this study. Data sharing is not applicable to this article.

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
