# Peer review of "Best Practices for the Use of High-Frequency Ultrasound to Guide Aesthetic Filler Injections—Part 2: Middle Third of the Face, Nose, and Tear Troughs"

_diagnostics, 2024, doi:10.3390/diagnostics14222544_

Round 1
Reviewer 1 Report
Comments and Suggestions for Authors
The submitted manuscript presents personal experience and exemplary guidelines for the injection of subcutaneous fillers in the midface. The authors use modern techniques to guide the needles to convenient and safe areas with full respect for the individual patient's anatomy. The manuscript is not a standard research paper in terms of content and structure, but a practical guide based on the authors' personal experience passed on to those interested in achieving safer, more precise, and successful interventions in aesthetic invasive cosmetics. The authors discuss their recommendations in great detail and confront complex sub-aspects with the state of the art. The manuscript has a high scientific standard and brings new insights contributing to the targeted field. I have no substantive comments on the text itself. I only ask the authors to add the specification of hyaluronic acid solutions to the Materials and Methods section. With this addition, I agree to publish the manuscript.
Author Response
Thank you very much for your positive feedback on our work; we’re truly pleased to know we’re on the right track. I’m not entirely clear on your suggestion regarding specifying the types of hyaluronic acid solutions. Are you suggesting that we mention specific brands? This guideline is actually intended to be applicable to any rheology or brand of hyaluronic acid, depending on the injector’s preference. The main goal is to guide injectors on using ultrasound to assist in injections, rather than recommending a specific brand or type of product for each treatment type.
Reviewer 2 Report
Comments and Suggestions for Authors
In this paper, we describe procedures using ultrasound using pictures and diagrams in an easy-to-understand manner for readers.
Although it is not very original, I think it is well described based on science.
However, I wish a video had been attached.
Author Response
Comment 1: In this paper, we describe procedures using ultrasound using pictures and diagrams in an easy-to-understand manner for readers.
Although it is not very original, I think it is well described based on science.
However, I wish a video had been attached.
Response 1: Thank you for your thoughtful feedback and for recognizing the clarity and scientific grounding of our descriptions. We appreciate your suggestion to include a video to further enhance the understanding of the procedures.
After careful consideration, we believe that the images and diagrams provided effectively illustrate the techniques and details necessary for readers. Our goal was to ensure that the static visuals are sufficiently comprehensive to convey the nuances of the procedures. We hope that these will serve our audience well, even in the absence of a video.
Thank you again for your constructive comments, which have been invaluable in refining our work.
Reviewer 3 Report
Comments and Suggestions for Authors
Every year, new techniques for addressing age-related changes are added to the fields of plastic surgery and aesthetic medicine. Experts can do minimally invasive operations such filler injections, cosmetic filament implantation, and exposure to physiotherapeutic factors in addition to surgery thanks to a variety of technology.
The majority of cosmetologists inject drugs into the thickness of facial tissues in an uncontrollable, nearly blind manner without considering the patient's anatomical features or a clear definition of the filler's location, shape, size, and topography in relation to other facial structures. This is partly due to the lack of recommendations on the technology of introducing fillers (the volume of the injected substance, the depth of penetration of the needle or cannula, and the methods of administration).
In addition to the possibility of introducing hyaluronidase into the vessel lumen or into the nerve, avoiding the gel itself, there is a significant danger of harming the tissues of blood vessels, nerves, and other structures in the treated area when medications are administered in this manner. In the intervention region, swelling and pain are caused by an inflammatory response of the surrounding tissues.
In addition to identifying anatomical variations, ultrasound enables you to ascertain the location and thickness of muscles, arteries, veins, and glands. Furthermore, ultrasonography offers non-invasive visualization of the face's anatomical layers, including the skin's epidermis, dermis, and superficial fat packages; the superficial muscular-aponeurotic system (SMAS), a network of connective tissue that exists between the skin and muscles and includes fibrous and elastic components; and the bones.
Making suggestions on the method of applying fillers under ultrasound is therefore a pressing issue.
With a wealth of illustrative information that enables you to clearly visualize the anatomical structures of certain facial areas, instances of filler injections using needles or cannulas, various ultrasound frequencies, etc., this section of the suggestions is presented rationally.
The writers mostly used papers that were no more than five years old at the time of publication when formulating their suggestions.
The authors conclude by demonstrating the necessity of employing ultrasonography for facial cosmetic treatments in order to prevent unfavorable outcomes, such as vascular disasters; ultrasound makes it possible to observe neurovascular plexuses, closely spaced veins, etc. All of this will enable professionals to enhance the caliber of cosmetic operations and prevent unwanted side effects.
The work's adherence to ethical standards might not be crucial in this instance because these are methodological suggestions.
In summary, the work is unique, well-written, and has the potential to serve as a methodological foundation for cosmetologists.
Author Response
Thank you very much for your encouraging review. We appreciate your recognition of the uniqueness and methodological value of our work, as well as your insightful reflections on the role of ultrasound in enhancing the safety and precision of cosmetic procedures. Your comments reinforce our commitment to advancing practical, evidence-based guidance for practitioners.
We are grateful for your support and thoughtful feedback.
Round 2
Reviewer 1 Report
Comments and Suggestions for Authors
This answers the authors' response in Report 1: I understand your question and doubts about my comment. For scientific publications, the Materials and Methods section specifies all the materials used, in your case HA, for which you do not provide details. You are right that in this case it may be the choice of the injector’s preference and the result should be achieved within a wide range of material specifications, regardless of the brand. On the other hand, there could be guidance on what HA solution to apply, with what concentration, what viscosity (to be able to push it through the application needle at all), and what molecular weight, so it is not about the HA manufacturer or brand but solution properties. If you do not have such information available or do not want to be specific, it is possible to at least subjectively describe the solution as "more fluid", "honey-like", or viscous but comfortably dispensable with a hand-held syringe. Your work is excellent, and I don't ask for anything exceptional that is not part of a scientific publication using any of the material.